# Structural Effects of Magnetostrictive Materials on the Magnetoelectric Response of Particulate CZFO/NKNLS Composites

**DOI:** 10.3390/ma12071053

**Published:** 2019-03-30

**Authors:** Moon Hyeok Choi, Kyujin Ko, Su Chul Yang

**Affiliations:** Department of Chemical Engineering, Dong-A University, Busan 49315, Korea; ansurl4927@gmail.com (M.H.C.); rbwls0096@gmail.com (K.K.)

**Keywords:** structural effect, magnetostrictive powders, hysteretic magnetization, magnetoelectric voltage, optimal dc magnetic field, particulate composites, CZFO, NKNLS

## Abstract

In this study, magnetostrictive powders of CoFe_2_O_4_ (CFO) and Zn-substituted CoFe_2_O_4_ (CZFO, Zn = 0.1, 0.2) were synthesized in order to decrease the optimal dc magnetic field (*H*_opt._), which is required to obtain a reliable magnetoelectric (ME) voltage in a 3-0 type particulate composite system. The CFO powders were prepared as a reference via a typical solid solution process. In particular, two types of heterogeneous CZFO powders were prepared via a stepwise solid solution process. Porous-CFO and dense-CFO powders were synthesized by calcination in a box furnace without and with pelletizing, respectively. Then, heterogeneous structures of pCZFO and dCZFO powders were prepared by Zn-substitution on calcined powders of porous-CFO and dense-CFO, respectively. Compared to the CFO powders, the heterogeneous pCZFO and dCZFO powders exhibited maximal magnetic susceptibilities (*χ*_max_) at lower *H*_dc_ values below ±50 Oe and ±10 Oe, respectively. The Zn substitution effect on the *H*_dc_ shift was more dominant in dCZFO than in pCZFO. This might be because the Zn ion could not diffuse into the dense-CFO powder, resulting in a more heterogeneous structure inducing an effective exchange-spring effect. As a result, ME composites consisting of 0.948Na_0.5_K_0.5_NbO_3_–0.052LiSbO_3_ (NKNLS) with CFO, pCZFO, and dCZFO were found to exhibit *H*_opt._ = 966 Oe (NKNLS-CFO), *H*_opt._ = 689–828 Oe (NKNLS-pCZFO), and *H*_opt._ = 458–481 Oe (NKNLS-dCZFO), respectively. The low values of *H*_opt._ below 500 Oe indicate that the structure of magnetostrictive materials should be considered in order to obtain a minimal *H*_opt._ for high feasibility of ME composites.

## 1. Introduction

Since the year 2000, magnetoelectric (ME) response has been a topic of interest in the development of energy-harvesters, sensitive magnetic sensors, and magnetically driven memories, or magnetoelectric transducers [1,2,3]. The ME effect is a result of induced piezoelectric effect (electrical effect/mechanical) in a piezoelectric phase by strain transfer of the magnetostrictive effect (mechanical/magnetic) in a magnetostrictive phase [4,5,6,7,8].
(1)$$\text{ME effect}=\frac{\mathrm{electric}}{\mathrm{mechanical}}\times \frac{\mathrm{mechanical}}{\mathrm{magnetic}}$$


However, reliable ME voltage from 3-0 type particulate composites can only be obtained under an optimal dc magnetic field (*H*_opt._) on the order of over several thousand Oersteds (Oe), which is a serious drawback limiting practical ME applications [9,10]. According to previous studies on particulate ME composites, a maximum ME voltage (*α*_ME_) was obtained at high values of *H*_opt._ above 1000 Oe from various compositions of Pb(Zr_0.52_Ti_0.48_)O_3_-Ni_0.8_Zn_0.2_Fe_2_O_4_ (*α*_ME_ = 54.4 mV/cm·Oe at *H*_opt._ = 1000 Oe), BaTiO_3_-Co_0.6_Zn_0.4_Fe_1.7_Mn_0.3_O_4_ (*α*_ME_ = 73 mV/cm·Oe at *H*_opt._ > 2000 Oe), BaTiO_3_-CoFe_2_O_4_ (*α*_ME_ = 17.04 mV/cm·Oe at *H*_opt._ > 15000 Oe), Ba_0.85_Ca_0.15_Ti_0.9_Zr_0.1_O_3_-CoFe_2_O_4_ (*α*_ME_ = 1.028 mV/cm·Oe at *H*_opt._ > 8000 Oe), and Na_0.5_Bi_0.5_TiO_3_-CoFe_2_O_4_ (*α*_ME_ = 0.42 mV/cm·Oe at *H*_opt._ >2500 Oe) [11,12,13,14,15]. Even though lower *H*_opt._ values of 500–1000 Oe were reported when investigating the size effect of magnetostrictive particles in BaTiO_3_-NiFe_1.98_O_4_ (*α*_ME_ = ~252 mV/cm·Oe at *H*_opt._ = 500–1000 Oe), the sintering temperature effect in Pb(Zr_0.52_Ti_0.48_)O_3_-NiCo_0.02_Cu_0.02_Mn_0.1_Fe_1.8_O_4_ (*α*_ME_ = 63 mV/cm·Oe at *H*_opt._ = 600 Oe), and the piezoelectric phase effect in Pb(Zr_0.52_Ti_0.48_)O_3_-Ni_1−*x*_Zn_*x*_Fe_2_O_4_ (*α*_ME_ = 190 mV/cm·Oe at *H*_opt._ = 800 Oe), there is still a need to decrease *H*_opt._ below 100 Oe for a high feasibility of particulate ME composite [16,17,18].

In this study, the structural effects of magnetostrictive materials on ME response was investigated in order to decrease *H*_opt._ values in a particulate ME composite system. In particular, magnetostrictive powders of CoFe_2_O_4_ (CFO), Zn-substituted porous-CFO (pCZFO) and Zn-substituted dense-CFO (dCZFO) were respectively prepared to explore structure-dependent hysteretic magnetizations. Then the *H*_opt._ shift in ME response was analyzed in particulate ME composites consisting of each magnetostrictive powder (CFO, pCZFO, and dCZFO) in a 0.948Na_0.5_K_0.5_NbO_3_–0.052LiSbO_3_ (NKNLS) piezoelectric matrix.

## 2. Experimental

Figure 1a–c shows a schematic diagram of the experimental procedure based on a solid-solution method to synthesize magnetostrictive powders of CFO, pCZFO, and dCZFO, respectively. As shown in Figure 1a, for preparation of CFO powders, Co_3_O_4_ (Sigma-Aldrich, Seoul, Korea, ≥99.5%) and Fe_2_O_3_ (Sigma-Aldrich, Seoul, Korea, ≥99.0%) powders were mixed by ball milling for 24 h. The well-mixed and fully dried powders were calcined at 1000 °C for 2 h. The calcined powders were ball-milled for 24 h and then sintered at 1200 °C for 2 h. After crushing and sieving of the sintered powders, CFO powders were selected with a particle size of 24–64 μm. As shown in Figure 1b,c, for preparation of pCZFO and dCZFO powders, Co_3_O_4_ (Sigma-Aldrich, Seoul, Korea, ≥99.5%) and Fe_2_O_3_ (Sigma-Aldrich, Seoul, Korea, ≥99.0%) powders were mixed by ball milling for 24 h. Then, the well-mixed and fully dried powders were calcined at 1000 °C for 2 h without and with pelletizing at 30 bar pressure, respectively. The calcined CFO powders exhibiting a porous structure (pCFO) and a dense structure (dCFO) were mixed with 0.1 and 0.2 molar ratio of ZnO powders (Sigma-Aldrich, Seoul, Korea, ≥99.0%), respectively. Then the mixed powders were sintered at 1200 °C for 2 h. After crushing and sieving of the sintered powders, pCZFO and dCZFO powders were selected with particle sizes of 24–64 μm.

ME composites were prepared with a 3-0 type particulate structure consisting of the magnetostrictive powders (CFO, pCZFO, and dCZFO, respectively) in a lead-free piezoelectric matrix of NKNLS. For preparation of NKNLS powders, K_2_CO_3_ (Sigma-Aldrich, Seoul, Korea, 99%), Na_2_CO_3_ (Sigma-Aldrich, Seoul, Korea, 99.5%), Li_2_CO_3_ (Sigma-Aldrich, Seoul, Korea, 99%), Nb_2_O_5_ (Sigma-Aldrich, Seoul, Korea, 99.9%), and Sb_2_O_5_ (Sigma-Aldrich, Seoul, Korea, 99%) powders were mixed by ball milling for 24 h. Then, the well-mixed and fully dried powders were calcined at 880 °C for 2 h. After sintering at 1050 °C for 2 h of CFO-NKNLS, pCZFO-NKNLS, and dCZFO-NKNLS pellets with a magnetostrictive/piezoelectric weight ratio of 0.1, disk-type ME composites were prepared with a thickness of 1 mm and a diameter of 13 mm. The ME composites were poled in silicone oil at room temperature by applying a dc field of 3 kV/mm for 30 min.

Crystal structures were investigated by X-ray diffraction (XRD; Miniflex600, RIGAKU, Tokyo, Japan) with CuK_α_ (λ = 1.5406 Å) radiation. The surface morphology was investigated by scanning electron microscopy (SEM; JEOL-6700F, Tokyo, Japan). Hysteretic magnetization curves were characterized by vibrating sample magnetometry (VSM; Model 7404, Lakeshore, CA, USA). Piezoelectric constants were measured by an APC YE 2730A d33 meter (APC Inc., Mackeyville, PA, USA). ME voltages were measured by applying an *H*_ac_ of 1 Oe at an off-resonance frequency, *f*, of 1 kHz using a lock-in amplifier (SR860, Stanford Research Systems Inc., Sunnyvale, CA, USA) [19,20]. As shown in Figure 1d, using the lock-in amplifier a calculated ac current was applied to a Helmholtz coil to induce an *H*_ac_ of 1 Oe with an off-resonance frequency of 1 kHz. Then, an *H*_dc_ of ±1000 Oe was applied to the ME samples using an electromagnet to obtain reliable ME voltages. Output ac voltage (*V*_ac_) from the ME samples was measured by the lock-in amplifier.

## 3. Results and Discussion

Crystal structures of the magnetostrictive CFO, pCZFO (Zn = 0.1, Zn = 0.2), and dCZFO (Zn = 0.1, Zn = 0.2) powders were investigated from XRD patterns. As shown in Figure 2a, all magnetostrictive powders were found to exhibit XRD peaks of (220), (311), (222), (400), (422), (511), and (440) representing a spinel structure of AB_2_O_4_ (JCPDS card No. 22-1086) [21,22]. Even though no noticeable peak shift in the XRD patterns was observed over a wide 2θ range after Zn substitution of 0.1 and 0.2 molar ratio on the porous-CFO and dense-CFO powders, a major shift of the (311) peak at 2θ = 35.5° towards a lower angle by Zn substitution was observed in the XRD patterns at a narrow 2θ range, as shown in Figure 2b. Bragg’s Law can be used to calculate a lattice constant using the equation:a^2^ = λ^2^(h^2^ + k^2^ + l^2^)^1/2^/4sin^2^θ(2)
where a is the lattice constant, λ is the wavelength of CuK_α_ radiation, and h, k, and l are the Miller indices. As the (311) peak shifts to a lower angle by Zn substitution, the lattice constant increases due to a decrease in the value of sin θ. With respect to the ionic radius, the pCZFO and dCZFO powders were found to exhibit an increased lattice constant compared to CFO powders because Zn^2+^ (0.82 Å) has a larger ionic radius than Co^2+^ (0.78 Å), which is replaced by Zn^2+^ [23,24,25].

In terms of Zn substitution in the porous-CFO and dense-CFO powders, magnetic properties of saturation magnetization (*M*_s_), remanent magnetization (*M*_r_), coercive field (*H*_c_), and magnetic susceptibility (*χ* = d*M*/d*H*) were investigated, as shown in Figure 3 and Table 1. Compared to the CFO powders, the pCZFO and dCZFO powders were found to exhibit enhanced *M*_s_ with decreased *H*_c_, as shown in Table 1. The enhanced values of *M*_s_ demonstrate that the addition of Zn^2+^ ions causes a migration of Fe^3+^ ions from a tetrahedral site to an octahedral site, which causes an increase of the total magnetic moment by reducing the net magnetic moment in the tetrahedral site. Furthermore, decreased values of *H*_c_ illustrate that grain growth by Zn substitution causes an increase of the domain wall number, resulting in large grain size, which requires less energy for spin rotation [26,27]. As shown in Figure 3b,e, stepped demagnetization behavior is shown by pCZFO with Zn = 0.2 and dCZFO with Zn = 0.1 and 0.2, which might be caused by the exchange-spring effect derived from the interplay of two uniquely characteristic phases [28,29,30]. From the result, it is noted that dCZFO possesses a sufficient exchange-spring effect based on high interaction between two magnetostrictive phases even though the Zn substitution of 0.1 is low in the dense-CFO powders. As shown in Figure 3c,f, the pCZFO and dCZFO powders were found to exhibit higher *χ*_max_ of 0.22–0.42 emu/g·Oe at lower values of *H*_dc_ below ±50 Oe, compared to *χ*_max_ of 0.05 emu/g·Oe at an *H*_dc_ below ±200 Oe from the CFO powders. In particular, the *χ*_max_ values of dCZFO were obtained at very low values of *H*_dc_ below ±10 Oe, which are induced by prominent stepped demagnetization behavior.

To investigate structure-dependent ME responses, particulate ME composites were prepared with compositions of CFO-NKNLS, pCZFO-NKNLS (Zn = 0.1, 0.2), and dCZFO-NKNLS (Zn = 0.1, 0.2). From the XRD patterns, as shown in Figure 4, perovskite (ABO_3_) and spinel (AB_2_O_4_) crystal structures were confirmed as piezoelectric and magnetostrictive phases, respectively. Even though sintering was conducted at 1050 °C for 2 h, all ME composites were found to exhibit stable crystal structures without any trace of secondary phase. In particular, a peak split at 2θ = 45–46° representing a tetragonal phase was maintained during the high temperature sintering. Therefore, the ME composites were found to exhibit a piezoelectric charge constant (*d*_33_) of 55–60 pC/N after sample poling.

From the particulate composites of CFO-NKNLS, pCZFO-NKNLS (Zn = 0.1, 0.2), and dCZFO-NKNLS (Zn = 0.1, 0.2), ME voltage (*α*_ME_) and *H*_opt._ were investigated while applying *H*_ac_ = 1 Oe at *f* = 1 kHz by sweeping *H*_dc_ of ±1000 Oe, as shown in Figure 5 and Table 2. The CFO-NKNLS composites were found to exhibit a maximum *α*_ME_ = 140 μV/cm·Oe at *H*_opt._ = 966 Oe. Even though a decreased *H*_opt._ value of 689–828 Oe was obtained from pCZFO-NKNLS as shown in Figure 5a, there was not a sufficient *H*_opt._ shift due to its weak behavior of stepped demagnetization. On the other hand, the dCZFO-NKNLS composites were found to exhibit remarkable *H*_opt._ values of 458–481 Oe as shown in Figure 5b, which are lower *H*_opt._ values than any reported particulate ME composites so far. As a result, the structural effect of magnetostrictive powders on *H*_opt._ shift is clearly shown between the heterogeneous pCZFO and dCZFO powders. Although the obtained *H*_opt._ value of 458 Oe from dCZFO-NKNLS is higher than 100 Oe, this study can serve to minimize a required *H*_opt._ by complexation with previous studies for high feasibility of particulate ME composites.

## 4. Conclusions

In this study, magnetostrictive powders of CFO, pCZFO (Zn = 0.1, 0.2) and dCZFO (Zn = 0.1, 0.2) were prepared to produce low values of *H*_opt._, which is required to obtain a reliable ME voltage in a 3-0 type particulate composite system. Compared to the CFO powders (*χ*_max_ = 0.05 emu/g·Oe at *H*_dc_ below ±200 Oe), the pCZFO and dCZFO powders were found to exhibit higher *χ*_max_ of 0.22–0.42 emu/g·Oe at lower *H*_dc_ values below ±50 Oe and ±10 Oe, respectively. The NKNLS-based ME composites consisting of CFO, pCZFO, dCZFO, respectively were found to exhibit *H*_opt._ = 966 Oe (NKNLS-CFO), *H*_opt._ = 689–828 Oe (NKNLS-pCZFO), and *H*_opt._ = 458–481 Oe (NKNLS-dCZFO). The results illustrate that a low *H*_opt._ value of 458 Oe was obtained from the effective stepped demagnetization behavior of dCZFO (Zn = 0.2), which was induced by a structural effect in a heterogeneous magnetostrictive phase.

## Figures and Tables

**Figure 1 materials-12-01053-f001:**
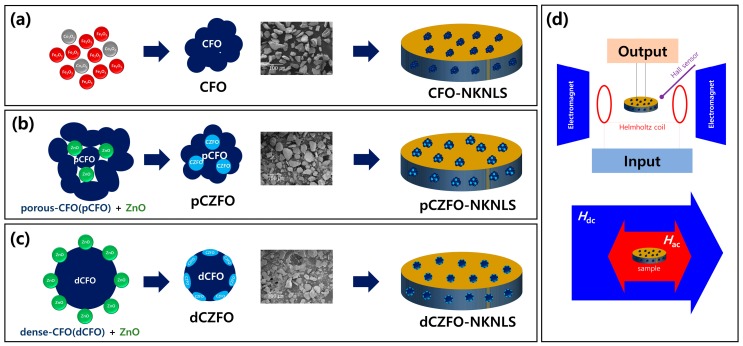
Schematic diagram of experimental procedure based on solid-solution synthesis for (**a**) CoFe_2_O_4_ (CFO), (**b**) Zn-substituted porous-CoFe_2_O_4_ (pCZFO), and (**c**) Zn-substituted dense-CoFe_2_O_4_ (dCZFO) powders. (**d**) Schematic diagram of magnetoelectric measurement set up.

**Figure 2 materials-12-01053-f002:**
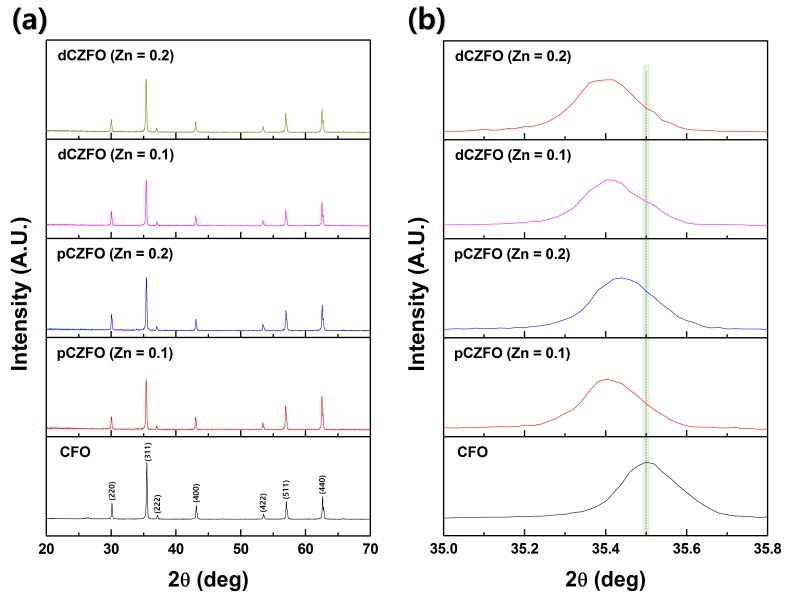
XRD patterns of CFO, pCZFO (Zn = 0.1, 0.2) and dCZFO (Zn = 0.1, 0.2) powders; (**a**) wide range 2θ of 20–70° and (**b**) narrow range 2θ of 35.0–35.8°.

**Figure 3 materials-12-01053-f003:**
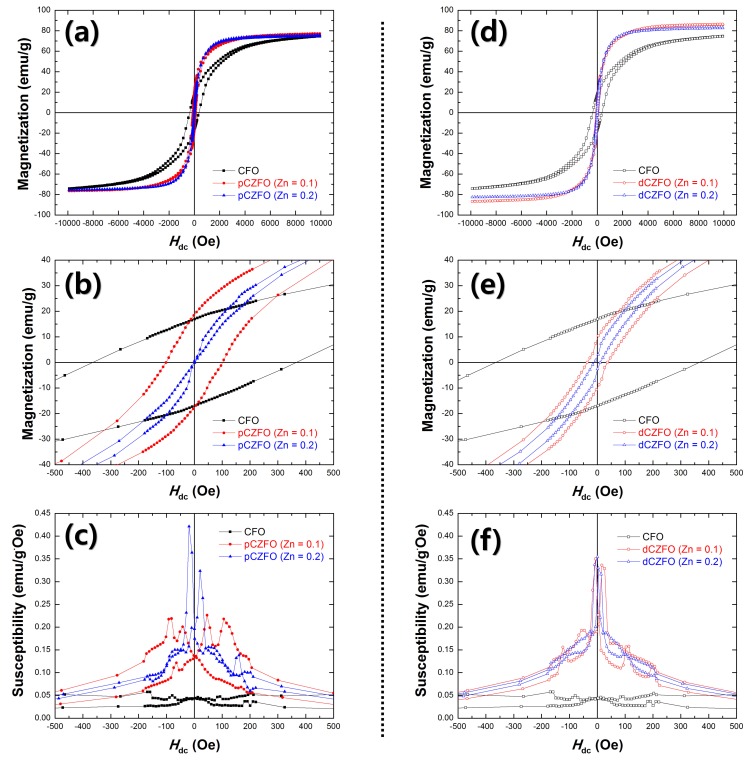
Hysteretic magnetization curves at (**a**,**d**) a wide *H*_dc_ range of ±10 kOe and (**b**,**e**) a narrow *H*_dc_ range of ±1 kOe, (**c**,**f**) magnetic susceptibilities (*χ*) of CFO, pCZFO (Zn = 0.1, 0.2) and dCZFO (Zn = 0.1, 0.2) powders.

**Figure 4 materials-12-01053-f004:**
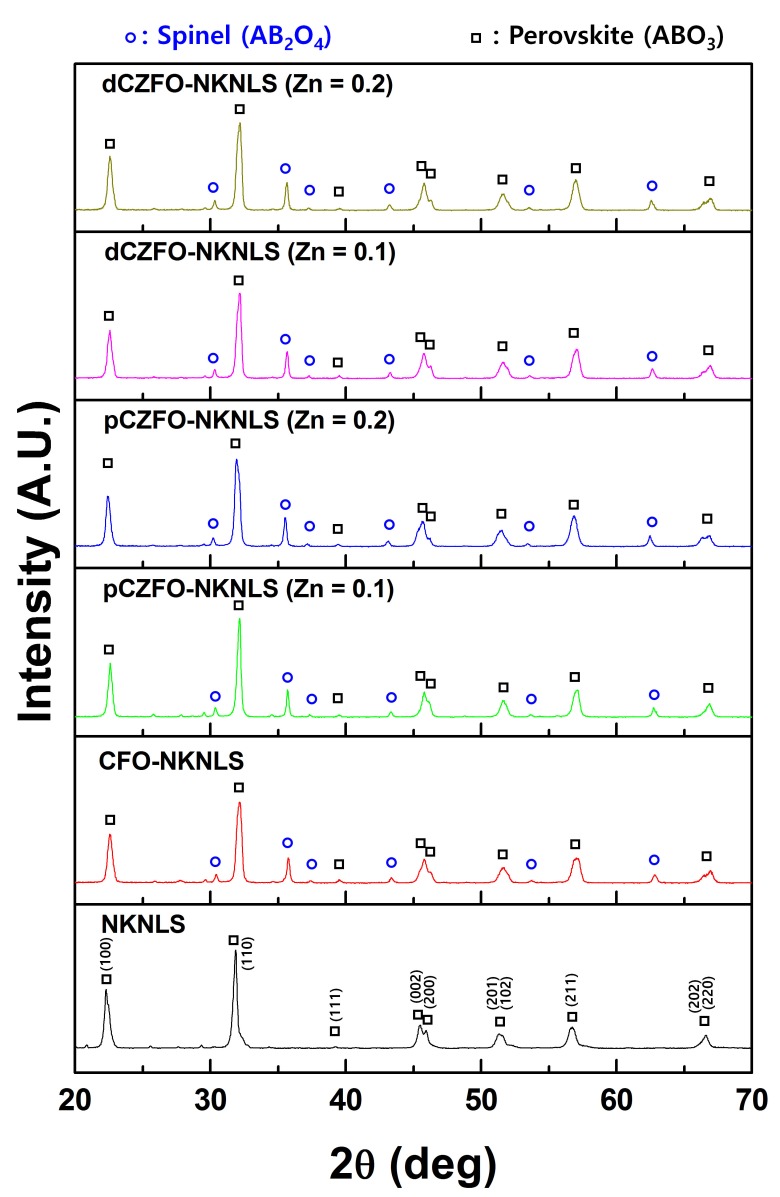
XRD patterns of magnetoelectric (ME) particulate composites consisting of a piezoelectric phase of NKNLS and magnetostrictive phases of CFO, pCZFO (Zn = 0.1, 0.2) and dCZFO (Zn = 0.1, 0.2).

**Figure 5 materials-12-01053-f005:**
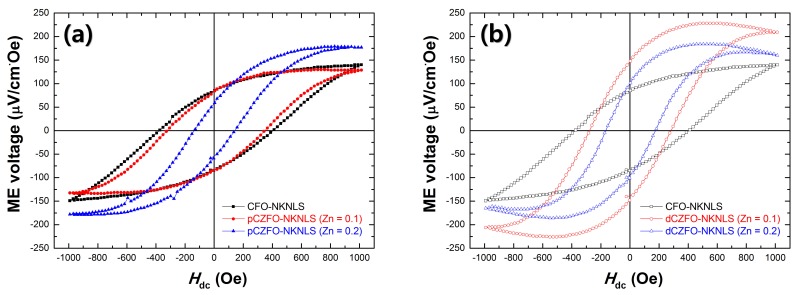
ME voltage of particulate composites consisting of a piezoelectric phase of NKNLS and magnetostrictive phases of (**a**) CFO and pCZFO (Zn = 0.1, 0.2) and (**b**) CFO and dCZFO (Zn = 0.1, 0.2).

**Table 1 materials-12-01053-t001:** Magnetic properties of CoFe_2_O_4_ (CFO), Zn-substituted porous-CoFe_2_O_4_ (pCZFO) and Zn-substituted dense-CoFe_2_O_4_ (dCZFO) powders; saturation magnetization (*M*_s_), remanent magnetization (*M*_r_), coercive field (*H*_c_), and magnetic susceptibility (*χ* = d*M*/d*H*).

Magnetostrictive Powders	Zn Ratio	Saturation Magnetization	Remanant Magnetization	Coercive Field	Magnetic Susceptibility
		*M*_s_ (emu/g)	*M*_r_ (emu/g)	*H*_c_ (Oe)	*χ*_max_ (emu/g·Oe)
CFO	Zn = 0	74.5 ± 0.75	16.8 ± 0.17	366.2 ± 3.66	0.05
pCZFO	Zn = 0.1	77.1 ± 0.77	18.5 ± 0.19	101.6 ± 1.02	0.22
	Zn = 0.2	75.9 ± 0.76	0.5 ± 0.01	2.4 ± 0.02	0.42
dCZFO	Zn = 0.1	86.3 ± 0.86	9.3 ± 0.09	36.2 ± 0.36	0.34
	Zn = 0.2	82.6 ± 0.83	2.3 ± 0.02	10.8 ± 0.11	0.35

**Table 2 materials-12-01053-t002:** Magnetoelectric (ME) responses of CFO-NKNLS, pCZFO-NKNLS, and dCZFO-NKNLS composites; optimal magnetic field (*H*_opt._) and ME voltage (*α*_ME_).

Magnetoelectric Composites	Zn Ratio	Optimal Magnetic Field	Magnetoelectric Voltage
		*H*_opt._ (Oe)	*α*_ME_ (μV/cm·Oe)
CFO-NKNLS	Zn = 0	966	140 ± 21.0
pCZFO-NKNLS	Zn = 0.1	689	130 ± 19.5
	Zn = 0.2	828	179 ± 26.9
dCZFO-NKNLS	Zn = 0.1	481	228 ± 34.2
	Zn = 0.2	458	184 ± 27.6

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
