# Peer review of "Structural Effects of Magnetostrictive Materials on the Magnetoelectric Response of Particulate CZFO/NKNLS Composites"

_materials, 2019, doi:10.3390/ma12071053_

Round 1
Reviewer 1 Report
The manuscript deals with studies on the structural effect of magnetostrictive particles on magnetoelectric responses in particulate CZFO/NKNLS composites. The paper is indeed of practical interest for applications based on the magnetoelctric effect. However, it seeks major revision before further considerations. These revisions include:
- The introduction is too short, too broad, and does not actually reflect the state-of-the-art research done so far in this topic.
- The scope of the paper is mostly centered on the materials synthesis. Therefore, it would be valuable to include different kind of materials in the introduction and some comments about the ME effect of such materials. Also, why the composite structure shows unique ME properties?
- The quality of Fig. 1 should be improved. It is hard to read the legends in a very tiny font size.
- In the experimental section, what is the ball-to-powder ratio during the ball milling?
- The particles are within the micrometer scale – so why the authors call them sometimes nanoparticles? (i.e. lines 42, 59, 85, 91, 133 & in the abstract).
- It would be also of interest if the authors comment on the feasibility of these particulate ME composites as mentioned in the abstract (the last sentence).
Author Response
We are very grateful to reviewers for positive and constructive comments as well as valuable time to review this manuscript. We are also pleased to resubmit this revised manuscript for publication in Materials. We have tried our best to address all received comments in the revised submission. We are submitting a clean version with all changes accepted.

Reviewer 2 Report
Paper can be accepted after the following corrections:
1. Experimental setup should be clearly presented and described in details.
2. Figure 1 is not clear. Please present it accordingly to scientific standards.
3. At least rough estimation of measurement uncertainty should be provided.
4. Please use SI units (e.g. A/m instead of Oe).
5. Figure 3c should be clarified.
6. Conclusion should be connected with possible applications indicated in the introduction.
Author Response

(The authors gave the same response as above.)

Reviewer 3 Report
The paper overall would benefit from being read by a native English speaker, as the grammar is poor and in places causes issues to allow understanding of the research.
The paper overall is very brief, and would benefit from a broader introduction and more analysis and discussion of the results. the results are just presented and no real effort is given to provide a "deeper" analysis and discussion. The figures just show data and numbers pulled from them with some general statements. A deeper discussion and understanding of the results presented and where they "sit" with other published research is required
A few other points:
- the text on figure 1 is too small to read when printed out
- page 3 explain in more detail the ME voltage measurements, and how the ME coefficients are achieved
- for the XRD data how does the full width at half max (FWHM) change between the different samples, what does this show
- also for the m-CZFO xrd spectra is there a second peak within the data at the same position as the CFO, has the peaks been modelled etc, if this is a second peak, what does this mean
- the list of numbers given from the magnetisation data might be better as a table, as then they are easier to compare
- SEM/TEM of the CZFO samples would be useful to confirm that they have different structures
- can the authors explain the sentence "Therefore, the m-CZFO particles were found to exhibit more effective magnetic interactions inducing large strains..", why can they state this, can they provide more evidence for this
- Also the sentence starting "From steffect (?) of magnetostrictive particles..." further analysis and discussion is required to justify this statement
- the whole of the results and discussion test was less than a page (not including figures), figure 4 is only mentioned in the text and no analysis of the data is presented, does the CZFO peaks change position in the matrix, how does the FWHM change with the matrix etc Further analysis is needed if this figure is included
- how does the ME voltages presented compared to any other research in the literature? Are they good or bad?
Author Response

(The authors gave the same response as above.)

Round 2
Reviewer 1 Report
The authors have improved their manuscript and replied to the referees' comments accordingly.
Author Response
Comments and Suggestions for Authors
The authors have improved their manuscript and replied to the referees' comments accordingly.
We sincerely appreciate your effort for the review of this manuscript. Your comment was really helpful for this study.
Reviewer 2 Report
Paper was corrected and can be accepted in the present form.
Author Response
Comments and Suggestions for Authors
Paper was corrected and can be accepted in the present form.
We sincerely appreciate your effort for the review of this manuscript. Your comment was really helpful for this study.
Reviewer 3 Report
The paper is better but there are still some concerns with it:
- The newly written abstract does not really portray the paper, there is also a difference in font size between the two paragraphs and the English is poor in places.
- line 30, contains poor English and needs rewriting
- line 42, magnetostrictive is spelt wrong
- line 267 states "exchange spring effect derived from the interplay of two uniquely characteristic phases", what are these two phases, as they aren't mentioned anywhere, are they detected in the XRD data presented? If not how do you know you have two phases, need to show evidence.
- do these two phases have the same structure? need to provide much more detailed information on these two phases
- how is the susceptibility determined? Is it just taking the dM/dH from the hysteresis loops? If so why aren't they symmetric?
- also why are some of the fields positive and the others negative? can the authors explain this
- errors are needs on the values given in the two tables
